# Aggrephagy Deficiency in the Placenta: A New Pathogenesis of Preeclampsia

**DOI:** 10.3390/ijms22052432

**Published:** 2021-02-28

**Authors:** Akitoshi Nakashima, Tomoko Shima, Sayaka Tsuda, Aiko Aoki, Mihoko Kawaguchi, Atsushi Furuta, Ippei Yasuda, Satoshi Yoneda, Akemi Yamaki-Ushijima, Shi-Bin Cheng, Surendra Sharma, Shigeru Saito

**Affiliations:** 1Department of Obstetrics and Gynecology, University of Toyama, Toyama 9300194, Japan; shitoko@med.u-toyama.ac.jp (T.S.); syk3326jp@yahoo.co.jp (S.T.); aikoyuzu8829@yahoo.co.jp (A.A.); mk.sagamore.hill@gmail.com (M.K.); at.furuta523@gmail.com (A.F.); ippeiy@med.u-toyama.ac.jp (I.Y.); s812yone@med.u-toyama.ac.jp (S.Y.); au@med.u-toyama.ac.jp (A.Y.-U.); 2Department of Pediatrics, Women and Infants Hospital of Rhode Island, Warren Alpert Medical School of Brown University, Providence, RI 02905, USA; shibin_cheng@brown.edu (S.-B.C.); ssharma@wihri.org (S.S.); 3University of Toyama, Toyama 9308555, Japan; s30saito@med.u-toyama.ac.jp

**Keywords:** aggrephagy, aggresome, autophagy, endoplasmic reticulum stress, inflammation, placenta, preeclampsia, pregnancy, protein aggregation, transthyretin

## Abstract

Aggrephagy is defined as the selective degradation of aggregated proteins by autophagosomes. Protein aggregation in organs and cells has been highlighted as a cause of multiple diseases, including neurodegenerative diseases, cardiac failure, and renal failure. Aggregates could pose a hazard for cell survival. Cells exhibit three main mechanisms against the accumulation of aggregates: protein refolding by upregulation of chaperones, reduction of protein overload by translational inhibition, and protein degradation by the ubiquitin–proteasome and autophagy–lysosome systems. Deletion of autophagy-related genes reportedly contributes to intracellular protein aggregation in vivo. Some proteins recognized in aggregates in preeclamptic placentas include those involved in neurodegenerative diseases. As aggregates are derived both intracellularly and extracellularly, special endocytosis for extracellular aggregates also employs the autophagy machinery. In this review, we discuss how the deficiency of aggrephagy and/or macroautophagy leads to poor placentation, resulting in preeclampsia or fetal growth restriction.

## 1. Introduction

Protein aggregation and accumulation in organs have been highlighted as causes of multiple diseases, including neurodegenerative diseases, cardiac failure, and renal failure. Aggregate proteins, which accumulate in intracellular or extracellular portions, disrupt cell function, resulting in organ failure. In the field of reproduction, the pathological implication of aggregated proteins has been noted, especially preeclampsia, a major cause of maternal and perinatal morbidity and mortality. This syndrome affects both the mother’s and child’s health during pregnancy, as well as in later life. Thus, women with preeclampsia are more likely to have complications from diabetes, cardiovascular diseases, kidney failure, and hypertension [1]. Moreover, the offspring of preeclamptic mothers present a higher risk of developing diabetes, hypertension, hormonal dysregulation, as well as delayed development and sensorimotor reflex maturation [2]. Female offspring who become pregnant tend to develop preeclampsia at a 2.6-time higher rate than offspring born from non-preeclampsia mothers [3], revealing that preeclampsia effects can be observed from one generation to the next.

Preeclampsia was originally diagnosed as newly emerged hypertension and proteinuria in pregnant women after 20 weeks of gestation. The placental structure is completed after 16 weeks of gestation, and poor placentation and placental abnormalities are related to the occurrence of preeclampsia. Furthermore, the International Society for the Study of Hypertension in Pregnancy recently altered preeclampsia diagnostic criteria, including some organ failures, placenta, as well as kidney or liver, in the diagnosis [4]. As the placental size is closely related to fetal size, failure of placental development results in fetal growth restriction (FGR) and various disorders in the offspring. Multiple factors, including hypoxia, systemic inflammation, failure of tolerance against the fetal antigen, failure of autophagy, or an increase in antiangiogenic factors, have been identified as possible causes of preeclampsia [5]. Among them, the increase in antiangiogenic factors, soluble Fms-like tyrosine kinase-1 (sFlt1) and soluble endoglin, in the maternal circulation are the most frequently investigated both in preeclampsia clinically and in basic science. The ratio of sFlt1 to placental growth factor has been used to predict preterm preeclampsia [6]. Aspirin, known to increase the serum concentration of placental growth factor in pregnant women [7], prevents the development of preterm preeclampsia in women who are at a high risk of developing preeclampsia and take aspirin from the first trimester [8]. Administration of sFlt1 and soluble endoglin leads to the development of preeclampsia in pregnant rats [9]. In contrast to these factors, the newly emerging concept of placental protein aggregation is now gaining momentum in the pathophysiology of preeclampsia. In this review, we discuss the role of protein aggregation in preeclampsia.

## 2. A Two-Stage to a Four-Stage Model for Placental Development

The etiology of preeclampsia has been discussed based on the “two-stage model” of preeclampsia. This is the most well-known model for preeclampsia development. Preeclampsia, especially the severe type, is frequently complicated with poor placentation, which occurs in the first stage, and systemic endothelial dysfunction at the second stage, and exhibits various symptoms, typically maternal hypertension and proteinuria, in pregnant women. During placentation, trophoblasts arise from the trophectoderm, the outer layer of the blastocyst, and invade the decidualized endometrium under a low nutrient and hypoxic environment, followed by implantation [10]. Proliferative cytotrophoblasts (CTBs) form cell columns, stalks of villi, and differentiate into syncytiotrophoblasts (STBs) on the fetal side, or into extravillous trophoblasts (EVTs) on the uterine side [11]. 

In human placentas, villi are covered with a multinucleated STB layer, generated by the fusion of underlying CTBs. The fusion of STBs, also called syncytialization, is required to transfer fetal requirements, including oxygen, amino acids, and organic ions, from maternal blood into fetal capillaries. Meanwhile, EVTs invade and reach one-third of the depth of the myometrium through the endometrium under approximately 2% oxygen tension [12]. The 2% oxygen concentration is maintained in the placenta until 12 weeks of gestation by obstructing maternal blood flow with a trophoblastic plug. Subsequently, the trophoblastic plug comes off from the maternal spiral artery, maternal blood flow enters the placenta in high volume, and placental oxygen tension steeply soars to approximately 7% oxygen tension. During placental development, the activation of hypoxia-mediated autophagy supports EVT invasion owing to the energy supply [13]. EVTs are classified based on the direction of invasion: interstitial EVTs toward the stroma and endovascular EVTs toward uterine arteries [14]. Endovascular EVTs replace the tunica media of spiral arteries, where apoptosis is induced by uterine natural killer cells [15]. This is called vascular remodeling and results in a sufficient supply of blood to the placenta. These two functions, invasion and vascular remodeling, are fundamental for normal placentation and could be affected by intracellular and extracellular factors. As interstitial and endovascular EVTs, which possess paternal antigens, first contact maternal immune cells and stromal cells, various cytokines and chemokines, known to respond to cell–cell contact, are exposed to EVTs. Some cytokines or chemokines are involved in EVT dysfunctions, resulting in poor placentation. Furthermore, failure or delay in trophoblast differentiation could result in poor placentation. However, the role of protein aggregation in poor placentation needs to be elucidated. 

Following the two-stage model developed by C.W. Redman, the “four-stage model” was proposed and has been accepted worldwide [16]. This concept adopts new maternal immunological tolerization against fetal antigens before and after implantation. For maternal immunological tolerance to occur in humans, a clonal increase of decidual regulatory T cells (Treg), which could be expanded against fetal/paternal antigens, is gradually increased in normal pregnant women depending on the gestational age [17]; conversely, clonally expanded effector memory CD8^+^ T cells express programmed death-1 (PD-1), which delivers inhibitory signals via binding with PD-L1, during normal pregnancy [18]. In women with preeclampsia, increased clonal Treg and PD-1 expression in clonal CD8^+^ T cells was reportedly inhibited in decidual lymphocytes. These results indicated the failure of maternal tolerization in the four-stage model of preeclampsia. It remains unknown whether functional disruption in trophoblasts by protein aggregation activates maternal immune reactions or vice versa. 

## 3. Role of Autophagy in Placental Development 

Cellular and organismal homeostasis is closely associated with protein quality control, which is balanced by protein translation, folding, and degradation. This balance could be impaired by intracellular or extracellular stress. Misfolded proteins, if not correctly refolded, accumulate in cells, resulting in aggregates posing a threat to cell survival. As autophagy is involved in protein degradation to maintain cellular homeostasis, this process is initiated by several stimuli, including hypoxia, starvation, or rapamycin, and torin-1, a more specific inhibitor of the mammalian target of rapamycin (mTOR), or Tat-Beclin1, a Beclin-1 activator [19]. During autophagy, an isolation membrane emerges in the cytoplasm and elongates to surround organelles non-selectively. Then, a double-membrane structure, an autophagosome, is completed with the closure of the vesicle. Subsequently, autophagosomes fuse with a lysosome, forming a complex called autolysosomes, and degrade the inner contents by employing lysosomal hydrolases (Figure 1) [20]. This process is involved only in macroautophagy, while others are micro-autophagy or chaperone-mediated autophagy. As for the molecular mechanism underlying this pathway, Atg4B, a cysteine protease, produces microtubule-associated protein 1 light chain 3 (LC3)-I, an inactive form of LC3, and mediates the conversion of LC3-I to LC3-II, a phosphatidylethanolamine-conjugated form, and an active form of LC3 [21]. Moreover, Atg4B converts LC3-II to LC3-1. LC3-II and the Atg5-Atg12-Atg16L1 complex play an important role in the elongation and completion of the autophagosome [22]. Sequestosome1 (SQSTM1), also known as p62, is a specific substrate of the autophagy machinery and is involved in autophagosome formation. 

To date, two autophagy-deficient trophoblast cell lines have been established, including HTR-ATG4B^C74A^ cells and HchEpC1b-ATG4B^C74A^ cells, as well as from EVT cell lines, HTR8/SVneo and HchEpC1b cells; in contrast, their counterparts include HTR8-mStrawberry cells and HchEpC1b-mStrawberry cells, respectively [23]. The original cell lines were stably transfected with pMRX-IRES-puro-mStrawberry-ATG4B^C74A^, an ATG4B^C74A^ mutant expression vector that inhibits LC3 conversion, or pMRX-IRES-puro-mStrawberry, a control vector encoding red fluorescent protein. These autophagy-deficient cell lines have shown sustained SQSTM1 expression levels, as well as complete deletion of LC3-II conversion under hypoxia. As stated earlier, invasion and vascular remodeling are fundamental functions in EVTs for normal placentation, and these functions are diminished in autophagy-deficient cells under hypoxia. Thus, hypoxia-mediated autophagy plays a role in normal placentation. To further verify the role of autophagy in placentation, trophoblast-specific Atg7 conditional knockout (cKO) mice were established. Atg7^flox/flox^ blastocysts, infected with a lentiviral vector expressing Cre recombinase, are transduced into the uterus in pseudo-pregnant mice. As lentivirus infection is restricted to the trophectoderm, but not the inner cell mass, while Atg7 expression is deleted only in the placenta, but not in the fetus [24]. Atg7 is involved in the process of autophagy activation by mediating the conjugations of the Atg12-Atg5 complex, as well as those of LC3 and phosphatidylethanolamine [25]; this resulted in cKO mice that showed the failure of autophagy in placentas, accompanied with increased SQSTM1 and decreased LC3-II in the placenta [24]. As for the phenotype in the cKO mice, blood pressure was gradually elevated in dams during pregnancy, and placental growth was restricted in the cKO placentas when compared with the control. Histological analysis revealed that SQSTM1 was considerably accumulated in the spongiotrophoblast layer, resulting in a smaller area than that in the control placenta. Furthermore, apoptosis was increased in the cKO placental layer (Figure 2). As Atg7 protects cells against metabolic stress by inducing cell cycle arrest mediated via the p53-p21 axis [26], apoptosis mediated by Atg7 deficiency might restrict placental growth. SQSTM1 specifically binds to aggregated proteins and is therefore used as a marker of aggregated proteins [27]. Moreover, SQSTM1 accumulates in trophoblasts in the spongiotrophoblast layer of Atg7 cKO placentas, and similar results were observed in cKO placentas with ProteoStat^®^ staining, a specific rotor dye for detecting aggregated proteins [28]. The cKO placentas were implanted into pseudo-pregnant mice, which present normal autophagic functions. Thus, carrying autophagy-deficient placenta elicits preeclampsia-like phenotypes in autophagy-normal dams; i.e., autophagy is required for normal placental development.

## 4. Aggrephagy in General

Macroautophagy was originally identified as a physiological process via which autophagosomes degrade some organelles non-selectively, especially under conditions of starvation, to maintain cellular homeostasis [20]. Currently, several types of “selective” autophagy processes have been named after the target, including aggrephagy (targeting protein aggregates), allophagy (targeting allogeneic organelles; i.e., elimination of paternal mitochondrial DNA by autophagy), chromatophagy (targeting chromatin), ER-phagy (endoplasmic reticulum: ER), ferritinophagy (targeting iron-bound ferritin), lipophagy (targeting lipid droplets), lysophagy (targeting lysosomes), pexophagy (targeting peroxisome), mitophagy (targeting the mitochondria), or xenophagy (targeting pathogens) [29]. Among them, aggrephagy involves the selective degradation of aggregated proteins by autophagosomes [30]. Misfolded proteins, which are produced by mutations or incomplete translation, or aberrant proteins, which are damaged by oxidative or other stress, can fail to form intact protein complexes. Once the proteins are degraded, these dispensable proteins are assembled and transferred to aggresomes dependent on microtubules for degradation (Figure 3). As the accumulation of misfolded proteins poses a hazard to cells, molecular chaperones are involved in repairing misfolding to maintain protein quality control in cells. However, if the damaged proteins are beyond refolding, misfolded or aggregated proteins are forwarded to two protein degradation systems: the ubiquitin–proteasome system and autophagy–lysosome system. Conversely, protein translation is transcriptionally inhibited to prevent the overload of misfolded proteins in cells [31]. Although protein translation is activated by mTOR, the activation of mTOR is involved in autophagy inhibition. Ras homolog enriched in the brain (RHEB), which activates mTOR complex 1, inhibits the transportation of protein aggregates to aggresomes by dissociation of the dynein–aggregate complex [32]. Accordingly, RHEB increases the sensitivity to apoptosis induced by aggregates.

The role of autophagy has been well studied in the field of neurodegenerative diseases such as Alzheimer’s disease (AD), amyloidosis, Parkinson’s disease, and polyglutamine diseases [33]. In clinical settings, immunohistochemical analysis of SQSTM1 is often used to detect a deficiency of aggrephagy in tissues [34,35], as loss of SQSTM1 in the mouse model suppresses protein aggregation induced by autophagy-deficient neurons and livers [36]. The other autophagy receptor proteins, autophagy-linked FYVE protein (ALFY), a neighbor of BRCA1, and TAX1 binding protein 1, are also included in protein aggregates, substrates of which are polyubiquitinated with the K63 chain [30]. ALFY enhanced the degradation of α-synuclein and polyglutamine inclusions with Atg5 [37]. Moreover, ALFY is involved in aggrephagy but not in starvation-induced autophagy. Histone deacetylase-6 (HDAC6), which binds to the F-actin cytoskeleton and microtubules for cellular movement, also plays a central role in transferring polyubiquitinated misfolded proteins to aggresomes via dynein (Figure 3). HDAC6 deficient-cells were sensitized to apoptosis induced by misfolded protein accumulation [38]. Although HDAC6 is dispensable for autophagy activation, HDAC6 is indispensable in basal autophagy via autophagosome–lysosome fusion, a fundamental process of autophagy, resulting in the enhancement of protein aggregation via F-actin [39]. The aggresome, composed of insoluble ubiquitinated proteins, is located at the microtubule organizing center (MTOC), the peri-nuclei envelope (Figure 3). BAG cochaperone 3 also binds to dynein to transport heat shock protein 70 substrates to the aggresome [40]. Other players involved in aggrephagy include p97/VCP (valosin-containing protein), a ubiquitin-associated HSP-independent molecular chaperone, and ubiquilin-1, a chaperone protein. In fibroblasts derived from patients with diseases such as inclusion body myopathy, Paget disease of bone, and frontotemporal dementia, mutation of p97/VCP revealed the impairment of autophagosome maturation like HDAC6 [41], as well as ER-associated degradation (ERAD) [42]. Similar to p97/VCP1, ubiquilin-1 also interacts with aggresome formation and ERAD. As decreased ubiquilin-1 levels have been reported in patients with AD, this protein may be related to the occurrence of late-onset AD [43].

Another possible role of autophagy in preventing neurodegenerative diseases via an aggrephagy-related mechanism is LC3-associated endocytosis, which is also called LANDO [44]. Deposition of amyloid β (Aβ) peptide in the central nervous system activates the inflammatory pathway, enhancing the progression of AD. LC3-associated endocytosis internalized the complex of Aβ and TREM2 (triggering receptor expressed on myeloid cells 2), a receptor of Aβ, via endocytosis in primary microglia derived from Rubicon (RUN and cysteine-rich domain-containing beclin1 interacting protein) knockout mice. This reduces Aβ deposition in the brain, as well as pro-inflammatory cytokine production by decreasing extracellular Aβ deposition. As LC3-associated endocytosis requires the process of LC3 maturation, Atg4, Atg5, Rubicon, and RavZ are indispensable for LC3-associated endocytosis, but FIP200 (FAK family kinase-interacting protein of 200 kDa), which is required for macroautophagy, is dispensable for LC3-associated endocytosis. In terms of more general extracellular targets, chaperone- and receptor-mediated extracellular protein degradation (CRED) is involved in reducing extracellular misfolded proteins [45]. Clusterin, which selectively binds to the misfolded protein as an extracellular chaperone, undertakes endocytosis via the heparan sulfate receptor, a proteoglycan, on the cell surface, delivering the cargo to the lysosome. As clusterin binds to Aβ, CRED may play an important role in the occurrence or progression of AD. Collectively, LC3-associated endocytosis and CRED might prevent the growth of misfolded proteins, which could be exaggerated by heat shock stress, oxidative stress, or inflammation in the bloodstream.

## 5. Aggrephagy in Placentas

Transthyretin (TTR), a protein that binds to thyroxine and retinol, which mainly forms a homotetramer, constitutes cytotoxic amyloid fibrils when pathological mutated TTR is produced in the liver, choroid plexus of the brain, and retina. TTR amyloidosis has been reported in a group of 25 diseases, including AD, owing to TTR deposition in the brain. An in vitro study showed that even wild-type TTR monomers rapidly assembled and formed small aggregates, resulting in cell death in tissue culture [46]. In serum samples from women with preeclampsia, surface-enhanced laser desorption ionization-time-of-flight, known as SELDI-TOF, revealed that TTR monomers in serum were significantly decreased during preeclampsia when compared with that in normal pregnant women [47]. The reduced serum TTR can be attributed to the deposition of TTR in preeclamptic placentas, in which the aggregated TTR exists as nanoparticles [48]. Furthermore, autophagy plays a role in eliminating nanoparticles in trophoblasts [49]. In addition, the TTR aggregates eluted from preeclampsia sera induced preeclampsia-like features, hypertension, and proteinuria in interleukin (IL)-10 knockout mice. Aggregated proteins, including Aβ precursor protein, can be detected in urine samples, as well as in placentas from women with preeclampsia [50]. Proteome analysis revealed differential protein profiles in the detergent-insoluble protein fraction between normal pregnancy and preeclampsia placentas [51]. The detergent-insoluble proteins included more endoglin, whose soluble forms are upregulated in the preeclampsia serum, and less vimentin, which is a structural protein of the aggresome. Thus, aggregates are present, but aggresome formation might be impaired in preeclamptic placentas.

A recent study focused on the pregnancy zona protein (PZP), which has an immunosuppressive effect and is produced by the placenta or leukocytes [52,53,54], reportedly inhibiting protein aggregation, including Aβ [55]. The concentration of PZP in plasma, also known as a protease inhibitor, peaks at approximately 3 mg/mL maximally, at the beginning of the 3rd trimester during pregnancy [56]; conversely, that in the non-pregnant population is approximately < 0.03 mg/mL. As mentioned before, clusterin inhibits protein aggregation, and haptoglobin and α2-macroglobulin demonstrate functions similar to clusterin. As PZP is a protein homologous to α2-macroglobulin, and these proteins, except PZP, are not increase during normal pregnancy [57,58], PZP might mediate endocytosis of aggregated proteins during pregnancy, like clusterin. In contrast, it has been reported that serum concentrations of clusterin are significantly upregulated in women with preeclampsia, compared with normal pregnant women, indicating the increase in aggregated proteins in the serum of women with preeclampsia [59].

For protein aggregates in preeclamptic placentas, SQSTM1 accumulation was observed in the syncytiotrophoblast layer, accompanied by the downregulation of lysosomal proteins [28]. Indeed, protein aggregates in sera from women with preeclampsia are considerably captured into HchEpC1b-ATG4B^C74A^ cells, the autophagy-deficient trophoblast cell line, when compared with their counterparts. Furthermore, this phenomenon was observed in Atg7 cKO placentas. Thus, aggrephagy deficiency, including macroautophagy deficiency, contributes to the deposition of protein aggregates in preeclamptic placentas [5]. In terms of the underlying molecular mechanism, downregulation of transcriptional factor EB (TFEB) is responsible for lysosomal dysfunction-mediated autophagy deficiency [28]. *Tfeb*, the basic Helix-Loop-Helix-Zipper family member, is well-known to centrally regulate autophagy and lysosomal biogenesis [46] and is essential for normal placentation. Deficiency of *Tfeb* results in embryonic lethality as the vasculature in the labyrinth layer of the placenta fails to grow owing to the loss of vascular endothelial growth factor [47]. In vitro studies have revealed that TFEB is downregulated by hypoxia in primary trophoblasts, and transactivation of TFEB, which moves into nuclei, is completely abolished by treatment with bafilomycin A1, an activator of TFEB, in the HchEpC1b-ATG4B^C74A^ cells. Inhibition of TFEB is retained by the hyperactivation of mTOR. The serum level of mTOR complex1, which is highly expressed in preeclamptic placentas, might be extremely useful for predicting hypertension, but not proteinuria, in women with preeclampsia between 24 and 28 weeks of gestation [60]. This is consistent with the findings of Atg7 cKO placentas, inducing hypertension and poor placentation, but not proteinuria in dams. TFEB downregulation was observed in two independent placenta-specific Atg7 knockout mice, cKO placentas, and labyrinth layer-specific Atg7 knockout placentas, accompanied by increased SQSTM1 and decreased lysosomal biogenesis [61]. Thus, autophagy inhibition, including aggrephagy inhibition, mediates hypertension rather than proteinuria in women with preeclampsia [62]. Another mechanism for aggrephagy inhibition is the reduction of p97/VCP, which blocks autophagosome maturation, as mentioned previously. Reduced p97/VCP can be observed in preeclamptic placentas with ubiquitinated protein accumulation [63]. It is still unclear whether protein aggregates affect cellular functions in trophoblast, resulting in preeclampsia. Hereafter, it should be figured out that protein aggregates in trophoblasts cause preeclampsia. 

## 6. Protein Aggregation and ER Stress in Placentas

ER stress has been identified in placentas presenting FGR, predominantly in FGR accompanied with preeclampsia, further eliciting placental growth restriction via translational inhibition [64]. Histological analysis of the human preeclamptic placenta revealed that cisternae of the ER are more dilated and the amorphous proteinaceous precipitates are considerably filled, which are signs of ER stress [65]. Repeated exposure to tunicamycin, an ER stress inducer, increased the expression of DNA damage-inducible transcript 3 (DDIT3, also known as CHOP), inhibiting placental and fetal growth [66]. The placental structure is aberrant, narrowing the vasculature in the labyrinth layer and decreasing the influx of maternal blood. The observed increase in DDIT3 expression in placentas with ER stress is consistent with that of human placentas with preeclampsia accompanied by FGR. Paradoxically, inositol-requiring enzyme-1 (IRE1), an ER-resident protein, is required for placental development; IRE-1 is activated in the placenta, but not in the fetus [67]. As observed in tunicamycin-treated placentas, the vascular structure in the labyrinth layer was disrupted in IRE1-knockout mice, accompanied by a reduction in vascular endothelial growth factor-A. Thus, moderate ER stress is required for normal placental development, low ER stress leads to miscarriage, and excessive stress affects placental growth, resulting in FGR and/or preeclampsia.

Excessive ER stress induced by chemical ER stress inducers decreased lysosomal numbers in trophoblast cell lines, resulting in the accumulation of autophagosomes and a decrease in autolysosomes, a sign of suppressed autophagic flux [68]. Chloroquine, bafilomycin A1, or wortmannin, known autophagy inhibitors, also increased ER stress, as confirmed by the increased expression of heat shock protein family A member 5 [68]. Collectively, autophagy inhibition exaggerates ER stress in trophoblast cells and vice versa. As mentioned earlier, lysosomes are involved in the degradation of aggregated proteins endocytosed by CRED. The accumulation of intracellular and extracellular aggregated proteins may be enhanced by ER stress in trophoblasts. Moreover, the lysosome-autophagy system is involved in the degradation of intracellular and extracellular aggregates in trophoblasts. 

Accordingly, IRE1α is highly expressed in placentas from early-onset preeclampsia pregnancies [69]. As IRE1-deficient *Dictyostelium* cells indicate collapsed ER cisternae, tunicamycin-induced ER stress inhibited autophagosome assembly on the ER membrane, suggesting that IRE1 maintains the ER membrane as a platform for autophagosome assembly [70,71]. Furthermore, ER stress induces protein aggregates, including ubiquitin, in IRE1-deficient cells. As ER-phagy, which has an inhibitory effect on the accumulation of aggregation-prone proteins [72], is required for maintaining ER homeostasis, monitoring and controlling ER-phagy can be considered to regulate protein aggregation in the placenta. Although ER-phagy shares a common molecular mechanism with macroautophagy, the specific molecular mechanism of ER-phagy demonstrates that family with sequence similarity 134, member B (FAM134B), which preferentially localizes in the ER membrane, binds to LC3 to attract autophagosomes, and Atlastin2, a dynamin-superfamily GTPase, enhances the fragmentation of ER marked with FAM134B, which is engulfed by autophagosomes [73,74]. Thus, macroautophagy could prevent the accumulation of aggregated proteins along with ER-phagy for protein quality control in human placentas.

## 7. Conclusions and Future Directions; Immunological Aspects in Aggrephagy for Pregnant Women

Selective and non-selective degradation of aggregated proteins by autophagy, including aggrephagy, is required for maintaining homeostasis in the placenta. Similar to neurodegenerative diseases, excessive protein aggregates lead to the disruption of normal functions in trophoblasts, resulting in preeclampsia or FGR. Conversely, the mechanism through which misfolded proteins aggregate in cells would differ between the placenta and central nervous system, as neurodegenerative diseases gradually progress with age, but pregnancy-related diseases progress in less than a year. This suggests that a unique mechanism for protein aggregation exists in placentas with preeclampsia and FGR. From the viewpoint of systemic inflammation observed in women with preeclampsia, this is caused by sterile inflammation because the placenta does not exhibit bacterial infection. Hypoxia-reoxygenation stress increased ER stress in a choriocarcinoma cell line, BeWo cells, and severe hypoxia-induced pro-inflammatory cytokine production in primary trophoblasts via pyroptosis [69,75]. Autophagy deficiency in trophoblasts enhanced pyroptosis-mediated production of pro-inflammatory cytokines, as ubiquitinated inflammasomes, which are earmarked with SQSTM1, were degraded by the autophagy machinery [76]. Furthermore, autophagy can maintain immunosuppression via Treg cells and prevents inflammatory cell infiltration into various organs [77]. Endometrial endothelial cells induce pro-inflammatory cytokines via phagocytosis of apoptotic trophoblast cells [78], and phagocytosis of dying cells is also mediated by autophagy-related LC3-associated phagocytosis. LC3-associated phagocytosis is involved in the elimination of dying cells to prevent excessive pro-inflammatory cytokine production [79]. The other possibility is that fetuses and placentas, semi-allografts, require immune tolerance from the maternal immune system. As clonally expanded maternal Treg cells against paternal antigens, which are maintained until the following pregnancy in the same individual, are likely to inhibit the occurrence of preeclampsia [80], autophagy might be required for the maintenance of clonal Treg cells during pregnancy. Moreover, non-degradative functions mediated by the autophagy machinery are related to homeostasis in the placenta [81]. There remain several aspects to investigate autophagy in preeclampsia and FGR. Accordingly, new therapies for placenta-mediated diseases have been developed.

## Figures and Tables

**Figure 1 ijms-22-02432-f001:**
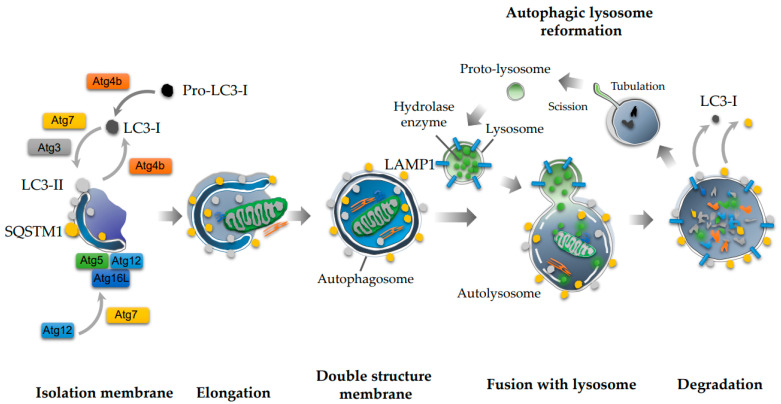
Autophagy cascade. Microtubule-associated protein 1 light chain 3 (LC3-I), which is generated from pro-LC3-I by Atg4B, is converted to LC3-II by Atg7 and Atg4B. Atg4B also mediates the conversion of LC3-II to LC3-I. An isolation membrane is elongated with LC3-II and Atg5-Atg12-Atg16L complex. The isolation membranes are closed, resulting in the autophagosome. Subsequently, the autophagosome fuses with lysosome, in which LAMP1 is expressed in the membrane, and digests the contents of the inner membrane. Lysosomes are also generated from the autolysosome by a recycling of protolysosomal membrane components.

**Figure 2 ijms-22-02432-f002:**
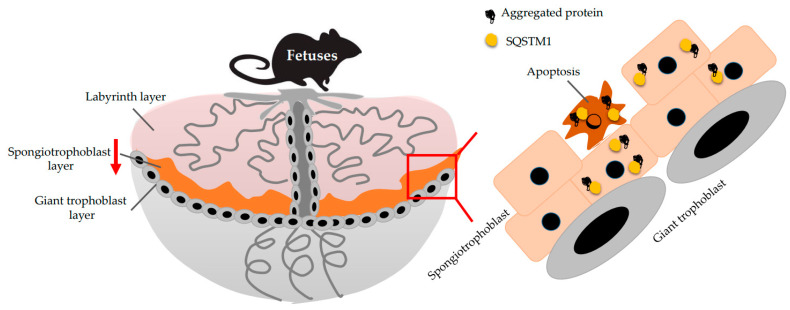
Aggregates in the Atg7 conditional knockout placenta. Atg7 deficiency in trophoblasts reduces the area of the spongiotrophoblast layer, compared with the control. Aggregated proteins accumulate in the area accompanied with SQSTM1 deposition, resulting in the increase of apoptosis.

**Figure 3 ijms-22-02432-f003:**
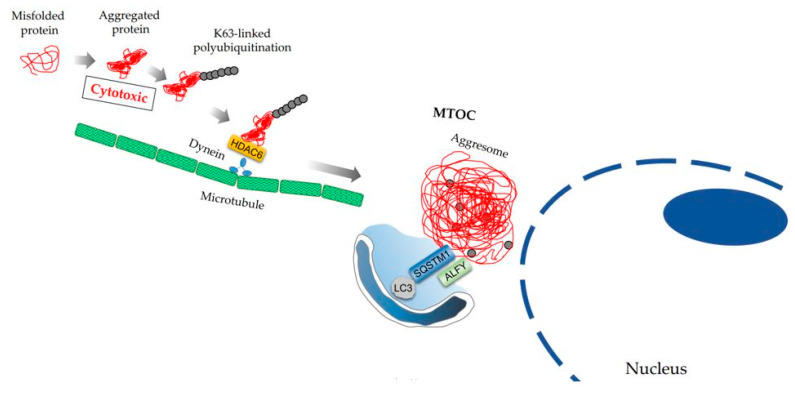
Transfer of aggregated proteins to aggresome in MTOC. Misfolded proteins build the cytotoxic aggregated protein. The aggregated protein, which is K63-polyubiquitinated, is trapped with HDAC6. The HDAC6 complex is transferred to the microtubule organizing center (MTOC) via dynein, resulting in aggresome. The autophagic adaptor proteins, SQSTM1 and ALFY, bind to aggresome to attract autophagosome membrane.

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
