# Peer review of "Aggrephagy Deficiency in the Placenta: A New Pathogenesis of Preeclampsia"

_ijms, 2021, doi:10.3390/ijms22052432_

Round 1

Reviewer 1 Report

  1. In view of the citated studies, accumulation of protein aggregates was obseverd in preeclampsia placenta. However, it is not clear whether the phenomenon is the cause or result of preeclampsia. If possible, please add data or  results of other studies that can explain the causal relationship between accumulation of protein aggregates and preeclampsia. If there is no study on this aspect, it is better to describe that the causal relationship is unclear.

  2. Even if some features of neurodegenerative disease, for example Alzheimer's disease, and preeclampsia show similarities, there are many differences between these diseases, especially the rate of disease progression and affected age. Therefore, it seems to be logical leap to designate preeclampsia as one of the protein aggregation disorders.

Author Response

Reviewer 1

Thank you for the reviewers’ comments. Following the reviewers’ comments, we carefully checked our manuscript as much as we could. We hope that the revision will fulfill reviewers’ requests.

Q 1. In view of the citated studies, accumulation of protein aggregates was observed in preeclampsia placenta. However, it is not clear whether the phenomenon is the cause or result of preeclampsia. If possible, please add data or results of other studies that can explain the causal relationship between accumulation of protein aggregates and preeclampsia. If there is no study on this aspect, it is better to describe that the causal relationship is unclear.

A1. Thanks for your comment. This is the crucial point that we have to clarify in our project. We’re now studying how much aggregated protein affects cellular mechanism, or which mechanism is vulnerable to aggregated proteins. Though there are a lot of reports about toxicity of aggregated proteins in central nervous system, there are few repots in this aspect in preeclamptic placentas. Therefore, we added the sentence as follows; “It is still unclear whether protein aggregates affect cellular functions in trophoblast, resulting in preeclampsia. Hereafter, it should be figured out that protein aggregates in trophoblasts cause preeclampsia.” (Line 320- 322).  

 Q2. Even if some features of neurodegenerative disease, for example Alzheimer's disease, and preeclampsia show similarities, there are many differences between these diseases, especially the rate of disease progression and affected age. Therefore, it seems to be logical leap to designate preeclampsia as one of the protein aggregation disorders.

A2. We recently propose the possibility that preeclampsia and some of neurodegenerative diseases have common mechanism for protein aggregation in each organ. Then we wrote “Regarding the correlation between women with preeclampsia and patients with AD, the common feature is misfolded protein aggregation in the organs. Women with preeclampsia present an increased risk for cardiac diseases and stroke, and are likely to develop dementia later in life [46].” (Line 255- 257). However, we haven’t reported the common molecular mechanism between them. Therefore, I deleted the sentences.

Reviewer 2 Report

The manuscript “Aggrephagy deficiency in the placenta: A new pathogenesis of preeclampsia” deals with the role of protein aggregation in the placental development and how the failure of the aggregates degradation machinery  may be involved in preeclampsia. The topic is intriguing and discussed exhaustively. The manuscript is well organized, but its readability is low because an overabundance of abbreviations and acronyms are present.  My suggestion is to limit their use to what is strictly necessary, to the names of enzymes or proteins, but not to the processes (LANDO or ERAD, for example). If the authors like to be more concise, they have to rephrase and simplify the sentences in order to explain clearly the research work discussed, as requested by the “review” form.

In conclusion,  I recommend the publication after these minor revisions.

Author Response

Q1. The manuscript “Aggrephagy deficiency in the placenta: A new pathogenesis of preeclampsia” deals with the role of protein aggregation in the placental development and how the failure of the aggregates degradation machinery may be involved in preeclampsia. The topic is intriguing and discussed exhaustively. The manuscript is well organized, but its readability is low because an overabundance of abbreviations and acronyms are present. My suggestion is to limit their use to what is strictly necessary, to the names of enzymes or proteins, but not to the processes (LANDO or ERAD, for example). If the authors like to be more concise, they have to rephrase and simplify the sentences in order to explain clearly the research work discussed, as requested by the “review” form. In conclusion, I recommend the publication after these minor revisions.

A1. Thanks for your evaluation. I totally agree with your advice. We tried to make our manuscript readable as much as we can. I hope our alternation makes it better for readers. But, CRED (chaperone- and receptor-mediated extracellular protein degradation) is a very long word, and we left it. We hope you understand our situation.
